# Physical- and Social-Based Rain Gauges—A Case Study on Urban Flood Detection

Vitor Y. Hossaki [1,2], Wilson F. M. S. Seron [3], Rogério G. Negri [1], Luciana R. Londe [2], Lívia R. Tomás [2], Roberta B. Bacelar [4], Sidgley C. Andrade [5] and Leonardo B. L. Santos [2,*]

1  Sciences and Technology Institute, São José dos Campos, State University (UNESP), São José dos Campos 12224-300, Brazil
2  National Center for Monitoring and Early Warning of Natural Disasters (Cemaden), São José dos Campos 12247-016, Brazil
3  Digital Media and Society Observatory, Federal University of São Paulo (UNIFESP), São José dos Campos 12231-280, Brazil
4  Department of Communication, Anhanguera College,  São José dos Campos 12236-660, Brazil
5  Federal University of Technology—Paraná (UTFPR), Toledo 85902-490, Brazil
*  Correspondence: santoslbl@gmail.com

**Abstract:** Floods are among the most frequent and costly rainfall-triggered disasters. In this context, geospatial content generated by non-professionals using geolocated systems offers the possibility of monitoring environmental events. This study shows a statistical correlation between in situsensors, radar, Twitter posts, and flooding events. Furthermore, we observed in this study that flooding-related keywords are statistically more significant on flooding days than on non-flooding days and reinforce that Twitter can be employed as a complementary data source for flood management systems.

**Keywords:** floods; Twitter; VGI; weather radar; rain gauge

## 1. Introduction

The Sustainable Development Goals (SDGs) are imperative for countries in the Global South, which are, in general, less developed and more impacted by climate change. Nevertheless, according to the "Sustainable Development Report 2022" [1], Brazil is among the countries with very low SDG commitment—those ranging from 0 to 40 per cent of governments' efforts toward the SDG Score. Based on the Sendai Framework for Disaster Risk Reduction 2015–2030 [2], the goal addressing Sustainable Cities (SDG 11) [1] includes access to safe transport systems; inclusive urbanization; reduction in the vulnerability of people affected by disasters; and, especially, more efficiency in the use of resources for adaptation to climate change and resilience to disasters. Concerning this goal, studies on urban floods support any action towards risk reduction for these kinds of disasters. These studies may consider characterization, impacts, tools, crowd-sensing, crowd-sourcing, and any other ways to contribute either to the understanding of or improvements in actions against flood impacts.

According to the 2021 International Panel on Climate Change (IPCC) [3], if global warming exceeds 1.5 °C, heat waves and extreme events, including floods, droughts, and intense rainfall, will be more frequent. In addition to increasing the incidence of these phenomena, climate change will enhance the duration and magnitude of events. Faced with these conditions, Early Warning Systems (EWS) are essential to reduce the risk of disasters.

An EWS comprises four essential elements: risk knowledge; monitoring and warning; dissemination and communication; and response capability. While some systems are aimed at risk identification [4] and decision making, other systems emphasize warning and response [5,6]. In the Flanders region of Belgium, the monitoring system is composed of a dense network of rain gauges associated with rain radar, and the forecasts are validated by long time series of flow and precipitation data. Cooperation with local communities and

authorities along with the EWS team has shown good results for flood forecasting [7]. The community is an essential factor for improving EWS performance [8]. Some models propose community information through the integration of geolocated social networks together with in situ and remote monitoring instruments. Ref. [9] suggested an architecture that collects and processes Voluntary Geographic Information (VGI) data from social networks (e.g., Twitter), collaborative maps (e.g., Open Street Map), and Wireless Sensor Networks (WSN); this system has indicated positive results for flood management.

However, the complexity and computational cost associated with filtering relevant data on social network in real time and integration with in situ and remote sensor data are challenges for researchers [10]. Furthermore, according to [11], social network activities are influenced by external factors and are spatially heterogeneous in complex urban areas.

Although social networks have become an opportunistic source of information for EWS scenarios where in situ and remote monitoring instruments are scarce or unavailable, another valuable source of crowdsourcing is Voluntary Geographic Information (VGI) [12]. OpenStreetMap is a well-known sort of project VGI (collaborative mapping) in which geographic information and features are voluntarily provided by non-expert citizens. Many initiatives adopt OpenStreetMap for mapping and analyzing humanitarian, crisis situations [13], road network quality [14], and land use/land cover [15] issues.

Another sort of VGI is the Citizen Observatories (crowd sensing) in which citizens contribute by providing information to authorities. For example, Ref. [16] developed a platform for citizens to share flood-related information or to provide information on dangerous situations. Ref. [17] developed and enriched the usability of a platform for citizens to share information regarding flooded areas and water levels in the riverbed. Nonetheless, it is worth highlighting that a common challenge of using opportunistic information sensing via social media networks (unconscious information) or volunteer information via VGI platforms (conscious information) for EWS scenarios is the lack of validity and uncertainty of information faced by official data [18].

In view of the presented discussions and motivations, we extracted Twitter data for analyzing the frequency of meteorological-related posts before flooding/non-flooding events and its correlation with physical–meteorological data, specifically, rain gauge and weather radar measures. A region comprising the lower part of the Tamanduateí river basin and data collected between January and March 2009 were considered in this study. The literature has demonstrated that Twitter data can be a valuable and cost-effective data source for improving situational awareness in a way that is not possible by conventional data sources [19].

This paper is organized as follows: Section 2 describes the study area and data sources and presents a brief discussion concerning the adopted statistical tests. Section 3 presents the obtained results and discussions, Lastly, Section 4 recaps the general features that permeate this paper, highlighting the main findings and methodological limitations.

## 2. Material and Methods

### 2.1. Study Area: The (Mega) City of São Paulo

The studied region comprises the lower part of the Tamanduateí river basin (Figure 1), Brazil. The basin's catchment area belongs to six municipalities, of which 115.3 km$^2$ are within the municipality of São Paulo. More than 80% of the Tamanduateí riverbed is impermeable, and floods along the riverbanks are recurring [20].

São Paulo is the largest city in the southern hemisphere and one of the largest conurbations in the world. It is one of the world's fastest-growing metropolitan populations [21], where almost 12.4 million people live in 1.521 km$^2$ (8152 in h/km$^2$). This municipality is considered critical in terms of disasters. In 2010, there were 674,329 people in areas at risk for floods, flash floods, and landslides [22].

Furthermore, this megacity is experiencing more frequent heavy rains and floods [23], which in turn can cause massive traffic problems [24]. In 2019, the city averaged 77 and 85 km of traffic jams in the morning and afternoon peak hours, respectively [25]. Ref. [26]

says that, in São Paulo, the opportunity costs arising from urban immobility can reach USD 1.4 billion every year.

### 2.2. Traditional/Physical and Social Data

Rainfall data were obtained from the National Center for Monitoring and Alerts for Natural Disasters (CEMADEN) database. The selected data were assigned to the local rain gauges 355030812A, 355030833A, and 355030857A (Figure 1—blue diamonds), whose influence on a neighborhood covers part of the study area. For convenience, the rain gauges are denoted by their last few characters (i.e., 812A, 833A, and 857A).

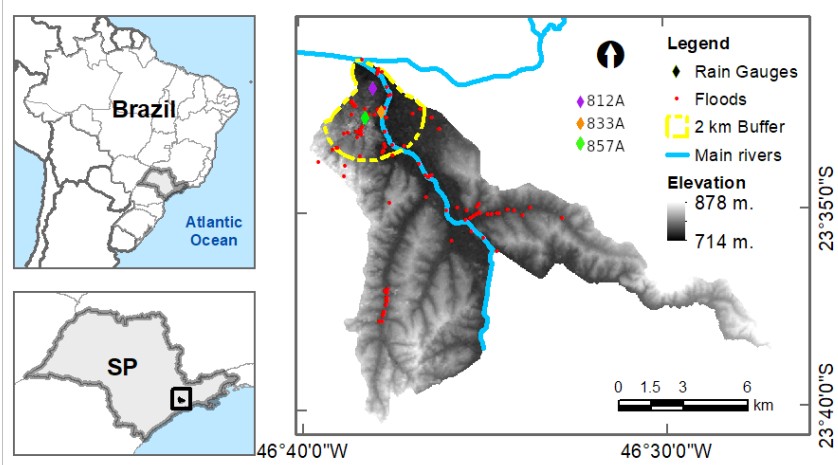

**Figure 1.** The lower part of the Tamanduateí Basin is depicted in the diagram, with diamonds indicating the locations of rain gauges, red markers indicating flood zones, and light blue representing the main hydrography. A 2-km buffer around the rain gauge locations is delineated by the yellow line, while the elevation of the study area is depicted by the black and white scale.

After the processing data (according to the flowchart—Figure 2), the registered values from the rain gauge were accumulated on a daily basis. The radar antenna was placed at "Pico do Couto" (S $22°27'51''$, W $43°17'50''$) at an altitude of 1771.94 m. It is an S-band radar with simple horizontal polarization. The radar has a range bin of 500 m and a scan frequency of 10 min, and data made available by the Department of Airspace Control (DECEA), with the spatial and temporal resolutions of 1 km and 10 min, respectively, expressed in terms of reflectivity (dB) into separation rate (mm/h), were converted according to the Marshall–Palmer relationship and then used in the calculation of accumulated values per day over the study area.

The data were obtained by the Emergency Management Center (CGE) [27], a government department in São Paulo city responsible for identifying and recording the geolocation of flooding incidents resulting from road obstructions. These records have the start and end times of road blocking and the street name and road intersection where a flood occurred. We used vector data with points representing traffic blocks due to floods generated by [28]. So, we selected the records from January to March 2019 that are within a 2 km radius of the rain gauges chosen for analysis (Figure 1). During the analyzed period, which coincides with the summer season in Brazil, and rainy season in the country's southeast, there has been a historically significant rise in the incidence of flooding and peaks in accumulated precipitation. Therefore, analyzing this period can provide valuable insights into the impacts of extreme weather events and their representation on social media. Heavy rainfall in São Paulo is generally defined by the Civil Defense as accumulations of more than 50 mm within a 24-h period. The situation is considered critical and may lead to floods. However, it is worth noting that local conditions, such as terrain and soil type, can also affect the amount of rainfall necessary to cause problems. There have been several instances of heavy precipitation, with accumulations exceeding 50 mm. For instance, on 2 March 2019, the rain gauge 812A recorded a peak of 56.12 mm/day, which resulted in five floods on that day.

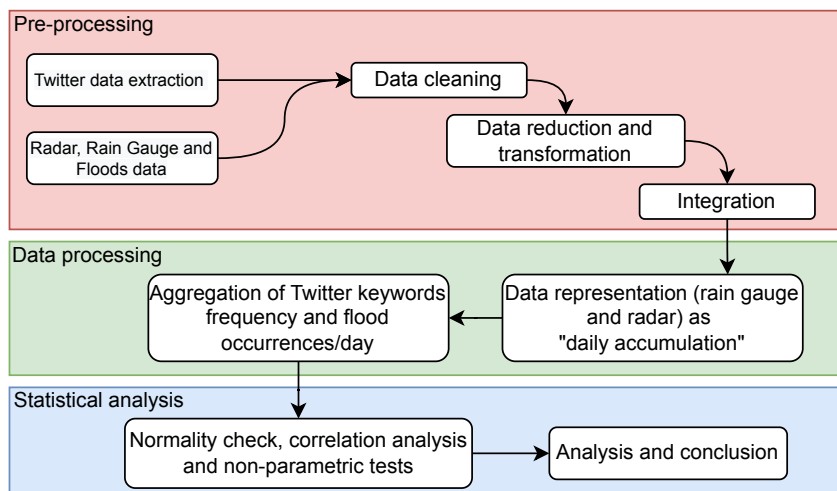

**Figure 2.** The flowchart illustrates all the treatment and processing steps undertaken for the analyzed data. The data were processed and analyzed using Python.

Tweets were extracted through Twitter's API Application Programming Interface); these data were collected in a spatial radius of 2000 m with the central point corresponding to the location of the CEMADEN Automatic Rain Gauges. The filtering process of Twitter data consisted of a list of words associated with rain and floods. To analyze the relevance of the words, we used a list of words consisting of rain-related keywords: "*chuva*", "*chove*", "*chuvoso*", "*chuvosa*", and "*chuvarada*". The terms used in the filtering process can be approximated as follows: chuva meaning rain, chove meaning it rains, chuvarada meaning downpour, and chuvoso/chuvosa meaning rainy. According to [11], such keywords are less spatially and temporally volatile than local and idiosyncratic terms. To further analysis, the tweets were separated into two temporal series that count the number of keywords detected in every filtered tweet, such as time series capable of revealing the social mirroring of rainfall patterns extracted from Twitter and allowing correlation analysis with rainfall data. In practice, this can evidence the use of Twitter as a supplementary data source for flood management, while the content of individual rain-related tweets can be used to improve the situational awareness.

The data used in our experiments were obtained through Twitter's Academy Research API. This API allows researchers access to the history of tweets, based on a coordinate and a radius, or by a set of keywords. In our paper, we chose the first option, as we were interested in a set of tweets that were posted within a 2 km radius [29] of the gauge stations.

*2.3. Statistical Tests*

Hypothesis tests are essential in proving statistical relations or divergences. Typically, the $\mathcal{H}$ is expressed by a null ($\mathcal{H}_0$) and an alternative ($\mathcal{H}_1$) hypothesis. While $\mathcal{H}_0$ implies no statistical significance, $\mathcal{H}_1$ states the opposite. Supposing a sample $\{x_i : i = 1, \ldots, n\}$ observed from a data space $\mathcal{A}$, we may infer about this sample in terms of $\mathcal{H}$ by identifying a critical region $C_\tau \subset \mathcal{A}$ that defines extreme situations and motivates the rejection of $\mathcal{H}_0$ under a probability $\alpha$ of making a wrong decision. The region $C_\tau$ is determined after imposing a threshold $\tau$, and the probability of occurring an event in $C_\tau$ is denoted by $p$.

Several procedures have been proposed in the literature to verify statistical hypotheses, for example, tests for single-sample tests, two paired or independent samples, correlations, concordance, and time series. Furthermore, these tests are categorized as parametric or non-parametric, where the latter does not consider premises about the population's distribution.

The Shapiro–Wilk [30], and Anderson–Darling [31] are examples of single-sample tests useful in identifying whether an observed sample was drawn from a population with a normal distribution. Furthermore, according to [32], the Shapiro–Wilk and Anderson–Darling tests are highlighted as the best tests to check data normality.

Let us suppose a sample $\{x_1, \ldots, x_n\}$. The Shapiro–Wilk test determines a statistic $SW$ that acts as a threshold for the rejection region:

$$SW = \frac{(\sum_{i=i}^{n} a_i x^{(i)})^2}{\sum_{i=i}^{n} (x_i - \overline{x})^2}, \tag{1}$$

where $\overline{x}$ is mean value computed from the sample set; $x^{(i)}$ is the $i$-th smaller element into the sample set; and $a_i$ is the $i$-th component of the vector that comes from $\frac{m^T V^{-1}}{\sqrt{m^T V^{-1} V^{-1} m}}$, considering $m = [m_1, \ldots, m_n]^T$ as the expected values with equivalent order and drawn from a standard normal distribution, and $V$ is the covariance matrix computed from $[m_1, \ldots, m_n]$. In order to compute the $p$-value, one needs to verify the theoretical distribution of $SW$ [33].

Similarly, the Anderson–Darling test will check the adherence between a sample set and an expected distribution, for example, the Gaussian distribution. The statistic regarding this test is presented in Equation (2):

$$AD = \left( -n - \sum_{i=1}^{n} \frac{2n-1}{n} (\ln F(x_i) + \ln(1 - F(x_{n+1-i}))) \right)^{\frac{1}{2}}, \tag{2}$$

where $F$ is the cumulative distribution function. Details about the $AD$ statistic distribution and $p$-value computing are found in [34].

Beyond the above-presented single-sample tests, the correlation and two independent samples tests help verify the relationship between variables and sets of observations. The Spearman's rank correlation [35] and Mann–Whitney $U$ [36] are examples of tests applied to such cases.

Concerning the Spearman's rank correlation, for a given set $\{(x_i, y_i) : i = 1, \ldots, n\}$ with paired observations according to the variables $X$ and $Y$, the following measure is computed:

$$\rho = 1 - \frac{6 \sum (R(x_i) - R(y_i))^2}{n(n^2 - 1)}, \, d_i = R(X_i) - R(Y_i), \tag{3}$$

where $R(\cdot)$ returns the rank of each observation according to the input variable. $\rho$ values closer to $-1$ or $1$ indicate intense direct or indirect correlation, respectively, between the compared variables [37]. In addition, the statistic $z = \rho\sqrt{\dfrac{n-2}{1-\rho^2}}$ follows a Student's $t$-distribution with $n-2$ degrees of freedom. Consequently, we may use such a relation to compute the $p$-value, as the probability of values equal or larger than $z$, and then check the significance of $\rho$ in rejecting the null hypothesis of no correlation between the variables [38].

Lastly, the Mann–Whitney $U$ test rises as a potential non-parametric alternative to compare two independent samples, allowing us to conclude if such samples come from the same population. Let us consider the two sets $\{x_1, \ldots, x_a\}$ and $\{y_1, \ldots, y_b\}$ with $a < b$, and the statistic test $U = \min\{U_a, U_b\}$, such that

$$\begin{aligned} U_a &= ab + \frac{a(a+1)}{2} - R_a \\ U_b &= ab + \frac{b(b+1)}{2} - R_b \end{aligned} \tag{4}$$

where $R_a$ and $R_b$ are the ranking sums regarding the element from the smaller and larger sets, respectively, after ordering its elements into a single set. When $a < 20$, a specific sampling distribution may be considered to define the assigned $p$-value [39]. Conversely, when $a \geq 20$, the statistic $U$ tends toward a normal distribution with parameters $\mu = \frac{ab}{2}$ and $\sigma = \sqrt{\frac{ab(a+b+1)}{12}}$, and a single-tailed $p$-value represents the probability of values equal or higher than $U$.

## 3. Results and Discussion

There were 166 traffic blocks due to floods in Tamanduateí basin in the period from January to March 2019, of which 89 were within a 2 km radius from the selected rain gauges. Furthermore, initially, we had 17,493, 89,114 and 79,896 tweets in the neighborhood of 812A, 833A, and 857A rain gauges, respectively. After filtering the data based on a list of keywords, were found 139, 548, and 492 tweets on the abovementioned rain gauges, respectively. The number of selected tweets was about 0.65% of the entire database. The graph depicting the data from the three analyzed points can be found in Figure 3.

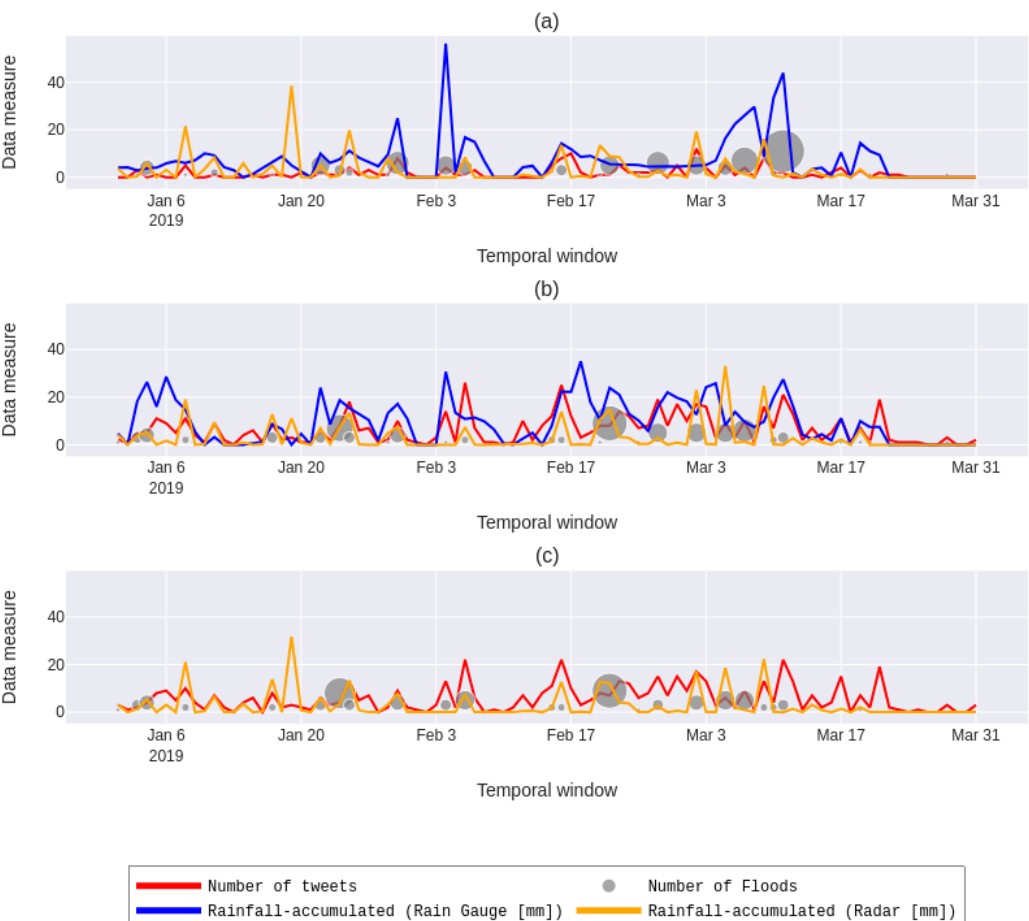

**Figure 3.** The graph illustrates the plot of three studied points (812A, 833A, and 857A) that match, respectively, with the (**a**–**c**) subplots. The y-axis displays the measurements from the rain gauge and radar in millimeters, as well as the absolute frequency of tweets and flooding. Meanwhile, the x-axis represents the time window under investigation. In order to enhance the visualization of the number of floods per day, the size of the markers varies proportionally to the amount of flooding that occurred on that day.

The normality of rain gauge measures and radar data were verified by applying the *SW* and *AD* tests. The 857A rain gauge data were not available on the CEMADEN website. According to Table 1, the performed tests indicate no sufficient evidence that the analyzed data sets are normally distributed. The *AD* statistic is far from the critical value at 5% significance ($\alpha = 0.05$), rejecting the null hypothesis that the data comes from a specific distribution. Furthermore, the *p*-value for the *SW* test is approximately zero, strengthening the conclusion that the analyzed data sets do not come from a normally distributed population.

**Table 1.** The tests were performed under a given significance ($\alpha$) of 5%. The critical value for *AD* test is 0.756. The prefixes R- and RG- stands for "RADAR" and "Rain Gauge", respectively.

|  |  | **R-812A** | **R-833A** | **R-857A** | **RG-812A** | **RG-833A** |
|---|---|---|---|---|---|---|
| *AD* | Statistic | 14.552 | 14.885 | 15.72 | 7.553 | 2.561 |
| *SW* | Statistic | 0.541 | 0.567 | 0.558 | 0.688 | 0.903 |
|  | *p*-value | 0.0 | 0.0 | 0.0 | 0.0 | $5.8 \times 10^{-6}$ |

Tweets, floods, and word frequency are discrete values that do not follow a specific distribution. Aligned with the statistical results of radar and rain gauge, the non-parametric Spearman's correlation was applied. Table 2 presents the Spearman's correlation ($\rho$) values with the *p*-value associated; there is a relevant statistical relationship between the data in the three rain gauges analyzed. Despite the absence of data for rain gauge 857A, it is possible observe the correlation between tweets, radar, and floods. Additionally, since all calculated *p*-values are inferior to the adopted 5% significance, it is identified statistical relationship between variables. The findings suggest that, in this case study, there is a weaker statistical relationship between extreme hydrological events and social media data when the volume of data is smaller. This is supported by the calculated p-values from the Spearman's correlation for the 812A being smaller than those for the 833A and 857A gauges.

**Table 2.** Spearman's correlation coefficient $\rho$ and respective *p*-value (in parenthesis).

|  | **812A** | **833A** | **857A** |
|---|---|---|---|
| Tweets and floods | $0.480 \ (1.64 \times 10^{-6})$ | $0.480 \ (1.66 \times 10^{-6})$ | $0.449 \ (8.73 \times 10^{-6})$ |
| Tweets and radar | $0.419 \ (3.90 \times 10^{-5})$ | $0.549 \ (1.99 \times 10^{-10})$ | $0.465 \ (3.79 \times 10^{-6})$ |
| Tweets and rain gauge | $0.525 \ (1.07 \times 10^{-7})$ | $0.656 \ (2.21 \times 10^{-10})$ | — |
| Floods and radar | $0.475 \ (2.21 \times 10^{-6})$ | $0.602 \ (3.24 \times 10^{-10})$ | $0.577 \ (2.49 \times 10^{-9})$ |
| Floods and rain gauge | $0.477 \ (1.93 \times 10^{-6})$ | $0.449 \ (8.95 \times 10^{-6})$ | — |
| Radar and rain gauge | $0.416 \ (4.47 \times 10^{-5})$ | $0.414 \ (4.77 \times 10^{-5})$ | — |

For further analysis of the relevance of geolocated tweets for flood detection, we counted the number of times that the keywords in filtered tweets appeared and performed a statistical comparison between the frequency of times that the keywords appeared on flood days and non-flood days. Figure 4 shows that the keyword frequency on flood days is superior to that on non-flood days. Considering that fewer tweets appear for 812A compared with the other points, the number of words on specific days can greatly deviate from the median (outliers) on non-flood days.

In other cases, such as 833A and 857A, it is evident that the words used on flood days are more relevant, since the majority values of the word frequency are above the median, and both cases are asymmetric positive boxplots. We have more variability in the data for the 833A and 857A points than in those for the 812A; despite the outliers in this last point, it seems that more tweets causes more variability, verified by the boxplots.

The Mann–Whitney *U* test was applied to verify the difference between flood days and non-flood days in the distribution of the (frequency of) keywords. The obtained *p*-values for 812A, 833A, and 857A were $1.267 \times 10^{-5}$, $4.200 \times 10^{-6}$, and $6.153 \times 10^{-7}$, respectively. These results make the differences already highlighted in Figure 4 evident. Furthermore, the computed *p*-values $\ll \alpha$ imply that the keyword distributions on flood days are significantly larger than those on non-flood days. The boxplot indicates that, at 812A, the number of tweets related to flooding was the lowest, and we observed the most outliers. This suggests that the rain gauge 812A generated less information from the social network and showed greater imprecision, as confirmed by the Mann–Whitney *U* tests.

Additionally, rain gauges 833A and 857A have approximately 3.5 times more flood-related tweets than 812A, as can be clearly seen in Figure 4.

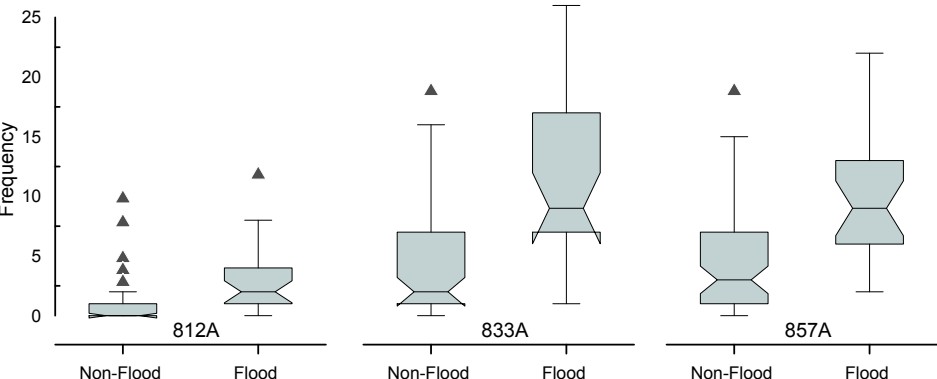

**Figure 4.** Boxplot with word distribution for flood days and non-flood days at the three points of the study area (812A, 833A, and 857A rain gauges). The notched box represents a 95% confidence interval, and the triangles are values out of the $\pm 1.5 \times$ interquartile range. When we refer to "frequency", we are specifically referring to the number of times a tweet keyword appears in the analyzed tweets, expressed as an absolute count. On the *x*-axis, the distribution of days with and without flooding is represented by the flooding and non-flooding days, respectively.

Considering this time window and geographic area, the general media highlighted 25 January, the date on which a heavy rain occurred, capable of causing a closure at Congonhas airport, in addition to being the anniversary of the capital Paulista. In the same way, other events mentioned in a specific way in the different vehicles (online, radio, television, and printed) were covered, with effort and space dedicated to each one compatible with the impact caused by the event. The information used by the vehicles to investigate the facts came from the CGE as well and from different sources, verified by the reporting teams of each one.

In addition to this type of coverage, we can identify other services/means of the communication between vehicles and their audience. The *Estadão portal*, for example, makes a tool called "It rained, stopped" available to its subscribers. Through the newspaper's website, it is possible to follow in real time which access roads in São Paulo are the most flooded. Another source of real-time information is the "listener reporter" system used by many radio stations, which open up their microphones to their listeners so that they can share how the traffic is in the city they are currently traveling through. Currently, humankind is surrounded by networks. Often, we are part of these networks ourselves, for example, with our personal relationships. Regardless of the domains from which the networks originate, structural information is only one part of the model of reality: each network, each node, and each connection within are subject to highly individual contextual circumstances. Such contextual information contains rich knowledge. They can enrich the structural data with explanations and provide access to semantics of structural connections. This contrasts with the classical methods of a network analysis, which indeed produces easily accessible and easily understandable results. However, so far, little attention has been paid to contextual information. This contextual information has yet to be made accessible to the algorithms of a network analysis.

The large availability of mobile computing technologies such as smartphones and tablets and the worldwide adoption of social applications such as Facebook and Twitter have allowed people to be continuously connected to the Internet. In this scenario, people act as social sensors, voluntarily providing data that capture their daily experiences from observations of the physical and online world.

## 4. Conclusions

This study makes a valuable contribution by rigorously demonstrating the statistical significance of tweets containing words related to hydrological extremes. By applying multiple tests of normality and non-parametric hypotheses, we have established a clear relationship between such tweets and extreme weather events. Our findings have important practical implications, particularly in the context of the significant economic and logistical losses caused by flooding. Furthermore, this research extends the existing literature on this topic and offers contributions for the development of technological tools to better address the challenges posed by flooding and its detrimental impacts.

Statistical analyses were carried out using a database with information about the number of traffic blocks due to floods, precipitation-based measures obtained by rain gauge and meteorological radar, and the number of tweets containing rain-related keywords. Preliminary verifications revealed that rain gauge and meteorological radar data are non-normally distributed, motivating the posterior analyses in a non-parametric fashion.

The highest correlations were found between the number of rain-related tweets and rain gauge measures. This result makes evident the potential of social data in predicting meteorological phenomena. Moreover, it is essential to highlight that the number of tweets is also correlated with radar measures and the number of flood events. In contrast, the weakest correlations were observed between rain gauge and radar measures, implying that such sources may be adopted as a complement.

Posterior comparative analyses (Figure 4) and statistical tests allowed for identifying that the number of rain-related tweets is different during/before flooding and non-flooding events. Furthermore, this difference suggests a local dependence.

About the tweet-collection process, Twitter's academic API has no period restrictions, but it is not possible to gain access to deleted or removed tweets as this violates Twitter's policy; however, as the search is carried out by a specific region and not by keywords, this limitation should not impact the research.

As Twitter data have provided positive evidence for flooding and have shown a positive correlation with meteorological data, further studies will be conducted using larger time series. Given the positive evidence for flooding from Twitter data and its positive correlation with meteorological data, it is necessary to conduct further studies with larger time series. Additionally, there are several areas that can be explored further. For example, we could apply supervised machine-learning models to develop more accurate alert systems and flood-forecasting tools. Additionally, by integrating meteorological data with social network data, we can better understand the spatial patterns of flooding and can develop more effective mitigation strategies.

**Author Contributions:** Conceptualization, L.R.L., S.C.A. and R.B.B.; methodology, validation, formal analysis and software, V.Y.H., R.G.N. and L.B.L.S.; resources and data curation, W.F.M.S.S. and L.R.T.; writing, editing and review, V.Y.H., L.R.L., L.B.L.S., S.C.A. and R.G.N.; supervision and project administration, L.B.L.S. All authors have read and agreed to the published version of the manuscript.

**Funding:** This research was funded by the São Paulo Research Foundation (FAPESP), grant 2021/01305-6, and National Council for Scientific and Technological Development (CNPq), grant 305220/2022-5. The APC was partially funded by Federal University of Technology—Paraná (UTFPR) and São Paulo State University (UNESP).

**Data Availability Statement:** The data sets of Twitter, the radar, the rain gauges, and the flood registry are available at https://github.com/vitor-yuichi/TweetFlood. Accessed date on 15 December 2022.

**Conflicts of Interest:** The authors declare no conflict of interest.

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
