# Peer review of "Physical- and Social-Based Rain Gauges—A Case Study on Urban Flood Detection"

_geosciences, doi:10.3390/geosciences13040111_

Round 1

Reviewer 1 Report

General comments:

1. The idea of using Twitter as a crowdsourcing data source is unique and interesting. Nonetheless, the present method is inapplicable for the forecasting of either high-intensity rainfalls or floods/flash floods. High EWS performance is almost impossible with crowdsourcing. However, it could be used for the geolocalization of extreme precipitation events and floods and likely for getting proxy data for rainfall intensities and the severity of floods. Also spatial data could be obtained the most affected location where disasters/floods frequently reoccur.

2. More case studies of extreme precipitation events should have been analyzed, not just a period of a few months. Select events of either high-intensities or high rainfall totals.

3. How were data obtained of the locations of the flood events? From OpenStreetMap? Please specify.

4. Compare (i) number of tweets (ii) radar intensities and (iii) ground precipitation intensities by rain gauges. Either in a table or on graphs. How accurately do the intensity and number of tweets reflect the precipitation intensity or the severity of floods and inundation? A map or a figure would be extremely useful and informative in this case. How are the results influenced by population density?

5. Specify radar data type. What type of radar was used? Conventional or polarization?

Specific comments:

There was no unit for the frequency of tweet keywords in Figure 2.

Please provide the English equivalents of the Portuguese keywords used in the study.

Was there any spatial difference among the three rain gauges based on the number or frequency of of tweet keywords? Could that be used to detect the spatial distribution of the severity of flood events or hydrological consequences of any sort?  

Following major revisions and improvements, the present manuscript will be useful not only in the fields of hydrology and hydrometeorology, but also in other environmental events and havocs that have an effect on large number of people.

Author Response

—————————————————

Dear Associate Editor and Reviewers, 

We would like to thank the editor and the reviewers for all their contributions towards improving our manuscript.

The adjustments in the manuscripts are highlighted according to the suggestions provided by Reviewer #1 (red), Reviewer #2 (blue), and Reviewer #3 (magenta).

We believe these improvements suffice to reach all the expectations and to provide high-quality discussions in the area.

Yours sincerely,

The authors

—————————————————

The idea of using Twitter as a crowdsourcing data source is unique and interesting. Nonetheless, the present method is inapplicable for the forecasting of either high-intensity rainfalls or floods/flash floods. High EWS performance is almost impossible with crowdsourcing. However, it could be used for the geolocalization of extreme precipitation events and floods and likely for getting proxy data for rainfall intensities and the severity of floods. Also, spatial data could be obtained in the most affected location where disasters/floods frequently reoccur.

R: Dear reviewer; We are delighted with your feedback. Thank you very much for the constructive comments and the time you spent reviewing our paper. Please, consider the modifications/inclusions highlighted in red in the paper.

Q2) More case studies of extreme precipitation events should have been analyzed, not just a period of a few months. Select events of either high intensities or high rainfall totals.

R: Many thanks for this comment. During the analyzed period, which coincides with the summer season in Brazil, and the rainy season in the country’s southeast, there has been a  historically significant rise in the incidence of flooding and peaks in accumulated precipitation. Therefore, analyzing this period can provide valuable insights into the impacts of extreme weather events and their representation on social media. Heavy rainfall in São Paulo is generally defined by the Civil Defense as accumulations of more than 50 millimeters within a 24-hour period. The situation is considered critical and may lead to floods. However, it's worth noting that local conditions, such as terrain and soil type, can also affect the amount of rainfall necessary to cause problems. There have been several instances of heavy precipitation, with accumulations exceeding 50 mm. For instance, on March 2, 2019, the rain gauge 812A recorded a peak of 56.12 mm/day, which resulted in five floods on that day. (Please, see lines 110–121).

Q3) How were data obtained of the locations of the flood events? From OpenStreetMap? Please specify.

R: Thank you for this question. The data were obtained by the Emergency Management Center (CGE), a government department in São Paulo city responsible for identifying and recording the geolocation of flooding incidents resulting from road obstructions. (Please, see lines 110–121).

Q4) Compare (i) number of tweets (ii) radar intensities and (iii) ground precipitation intensities by rain gauges. Either in a table or on graphs. How accurately do the intensity and number of tweets reflect the precipitation intensity or the severity of floods and inundation? A map or a figure would be extremely useful and informative in this case. How are the results influenced by population density?

R: Many thanks for this suggestion. We included new graphics (see new Figure 3) in the revised manuscript to compare the data concerning the rain gauges. Additionally, since the visual comparison is challenging, Spearman’s correlation indexes, box plots, and hypothesis tests were also used to support solid discussions. Statistical analysis by Mann-Whitney’s hypothesis tests and box plots indicates that the number of tweets related to hydrological extremes is a reliable indicator of flooding intensity. These analyses demonstrate that the words contained in tweets have higher statistical relevance on days with flooding events. This suggests that analyzing tweets can provide valuable insight into the impact of hydrological extremes, allowing for a better understanding of the spatial and temporal patterns of flooding events. (Please, see lines 233–238).

Q5) Specify radar data type. What type of radar was used? Conventional or polarization?

R: Thank you for this comment. The radar antenna is placed at “Pico do Couto” (S 22º27’51’’, W 43º17’50’’) at an altitude of 1771.94 m. It is an S-band radar with simple horizontal polarization. The radar has a range bin of 500 m and a scan frequency of 10 min. These details were included in the revised manuscript (please, see lines 97–99).

Q6) Specific comments:

 a) There was no unit for the frequency of tweet keywords in Figure 4.

R: We have included the dimensions of each measurement on the axes in the Figure 2 description.

b) Please provide the English equivalents of the Portuguese keywords used in the study.

R: We have included the English equivalents of these words in the manuscript."The words utilized in the filtering process can be approximated as: chuva | rain; chove | it rains; chuvarada | downpour; | (chuvoso/chuvosa) | rainy;” (Please, see lines 127–129).

 c) Was there any spatial difference among the three rain gauges based on the number or frequency of tweet keywords? Could that be used to detect the spatial distribution of the severity of flood events or hydrological consequences of any sort?  

R: The analysis revealed a notable difference in the 812A rain gauge. Only 139 tweets related to rain or flooding were identified after processing, compared to 548 and 492 tweets for the 833A and 857A rain gauges, respectively. Moreover, the 812A gauge showed a smaller difference between the frequency of relevant keywords and flooding compared to the other gauges. These findings suggest that, in this case study, there is a weaker statistical relationship between extreme hydrological events and social media data when the volume of data is smaller. This is supported by the calculated p-values from Spearman’s correlation, which were about 10 times smaller for the 812A gauge than for the 833A and 857A gauges. 

The above-presented discussion was added to the revised manuscript (Please, see lines 217-220).

Following major revisions and improvements, the present manuscript will be useful not only in the fields of hydrology and hydrometeorology but also in other environmental events and havocs that have an effect on a large number of people.

R: We are very thankful for your insightful comments and suggestions.

Reviewer 2 Report

Reviewers' comments:

This paper presents a case study on urban floods detection using physical and social-based rain gauges. It is an interesting paper; however, in my opinion, the paper can be accepted for publication after considering the following revisions as follows:

1.      Please clarify the relevance or novelty and contributions of this work. Please explain why the present study is more important compared to existing works in this direction. To capture the interest of readers, the main contribution and advantages of this work should be indicated.

2.      The description of Table 1 must be improved. It should be “The location of the study area”.

3.      In this article, the author chooses Twitter data for analyzing the frequency of meteorological-related posts. However, the author did not introduce it clearly about meteorological data preparation of these physical-meteorological data。Why choose the size of the 2000 m as spatial radius?

4.      The most relevant data-results should be summarized and demonstrated by a graph and a corresponding table. Additionally, please highlight the outliers in all the tables and graphs, where relevant.

5.      The methods are not correctly described and sufficiently informative to allow replication of the research. The methodology of this study should be described by a schematic and conceptual flowchart.

6.      From the Material and Methods, it is not clear enough how the Twitter can be employed as a complementary data source for flooding management systems. The description and the benefits of Twitter data are not sufficiently presented. It is difficult to make general conclusions on the method performance from the presented results.

7.      To highlight the relevance and reliability of these results, a comparison should be made with the flood sustainability maps obtained in other works. How far or close are the results of this study from other studies?

8.      Please briefly add future perspectives and further applied applications of this specific research work in the discussion section.

Author Response

—————————————————

Dear Associate Editor and Reviewers, 

We would like to thank the editor and the reviewers for all their contributions towards improving our manuscript.

The adjustments in the manuscripts are highlighted according to the suggestions provided by Reviewer #1 (red), Reviewer #2 (blue), and Reviewer #3 (magenta).

We believe these improvements suffice to reach all the expectations and to provide high-quality discussions in the area.

Yours sincerely,

The authors

—————————————————

This paper presents a case study on urban floods detection using physical and social-based rain gauges. It is an interesting paper; however, in my opinion, the paper can be accepted for publication after considering the following revisions as follows:

R: We are delighted with your feedback. Thank you very much for the constructive comments and the time spent reviewing our manuscript. Please, consider the modifications/inclusions highlighted in blue in the paper.

Q1) Please clarify the relevance or novelty and contributions of this work. Please explain why the present study is more important compared to existing works in this direction. To capture the interest of readers, the main contribution and advantages of this work should be indicated.

R: Thanks for pointing out this issue. This study makes a valuable contribution by rigorously demonstrating the statistical significance of tweets containing words related to hydrological extremes. By applying multiple tests of normality and non-parametric hypotheses, we have established a clear relationship between such tweets and extreme weather events. Our findings have important practical implications, particularly in the context of the significant economic and logistical losses caused by flooding. Furthermore, this research extends the existing literature on this topic and offers contributions for the development of technological tools to better address the challenges posed by flooding and its detrimental impacts. 

The above-presented discussion was added in the revised manuscript (Please, see lines 275–282).

Q2) The description of Table 1 must be improved. It should be “The location of the study area”.

R: Thank you for this suggestion. The description of Figure 1 was improved.

Q3) In this article, the author chooses Twitter data for analyzing the frequency of meteorological-related posts. However, the author did not introduce it clearly about meteorological data preparation of these physical-meteorological data. Why choose the size of the 2000 m as the spatial radius?

R: Many thanks for this question. The adopted radius is a classical approximation from the literature. We have included a reference [Chacon-Hurtado et al. (2017)] in the revised manuscript to support such a decision. (Please, see lines 139-140).

Q4) The most relevant data-results should be summarized and demonstrated by a graph and a corresponding table. Additionally, please highlight the outliers in all the tables and graphs, where relevant.

R: Thank you very much for this suggestion. We included new graphics (see new Figure 3) in the revised manuscript to compare the data concerning the rain gauges. 

The boxplot indicates that at 812A, the number of tweets related to flooding was the lowest, and we observed the highest number of outliers. This suggests that the rain gauge 812A generated less information from the social network and also showed greater imprecision, as confirmed by the Mann-Whitney tests. Additionally, rain gauges 833A and 857A have approximately 3.5 times more flood-related tweets than 812A, as can be clearly seen (Table 2). As a result, the distribution of outliers at these points is significantly smaller.

The above-presented discussion was added in the revised manuscript (Please, see lines 239–244).

Q5) The methods are not correctly described and sufficiently informative to allow replication of the research. The methodology of this study should be described by a schematic and conceptual flowchart.

R: Many thanks for this comment. We included a conceptual flowchart (see Figure 2) in the revised manuscript as further methodological details.

Q6) From the Material and Methods, it is not clear enough how the Twitter can be employed as a complementary data source for flooding management systems. The description and the benefits of Twitter data are not sufficiently presented. It is difficult to make general conclusions on the method performance from the presented results.

R: Thank you for this comment. We added sentences in the “Introduction” and "Material and Methods" sections to make clear the value of Twitter data (social media data) for flood management, which uses human perception of surroundings as a complementary data source. With regards to benefits, we have mentioned in the section Introduction “[...] social networks have become an opportunistic source of information for EWS scenarios where in situ and remote monitoring instruments are scarce or unavailable”. In addition to these scenarios, social media data (e.g. Twitter data) can also provide situational awareness in a way that is not possible by conventional data sources (pluviometers and weather radars). Please, see lines 66–69 and 132-136.

Follow the sentences added:

Introduction

The literature has demonstrated that Twitter data can be a valuable and cost-effective data source for improving situational awareness in a way that is not possible by conventional data sources (Huang and Xiao, 2015). Please see line 66–69

Huang, Q.; Xiao, Y. Geographic Situational Awareness: Mining Tweets for Disaster Preparedness, Emergency Response, Impact, and Recovery. ISPRS Int. J. Geo-Inf. 2015, 4, 1549-1568. https://doi.org/10.3390/ijgi4031549  

Material and Methods

Such as time series are capable of revealing the social mirroring of rainfall patterns extracted from Twitter and allowing correlation analysis with rainfall data. In practice, this can evidence the use of Twitter as a supplementary data source for flood management, while the content of individual rain-related tweets can be used to improve the situational awareness. 

Q7) To highlight the relevance and reliability of these results, a comparison should be made with the flood sustainability maps obtained in other works. How far or close are the results of this study from other studies? 

R: We thank the reviewer for this comment. Comparing long-time series of georeferenced tweets with susceptibility maps is an excellent idea for future studies. We have included it in the perspectives of this paper. We can not address it in this paper once we are interested in short-time periods from an early-warning system perspective. 

Q8) Please briefly add future perspectives and further applied applications of this specific research work in the discussion section.

R: Many thanks for this insightful suggestion. We included in the “discussion” section further considerations about the perspectives and applications of our research. (Please, see lines 301–308)

Passage added to the manuscript: “As Twitter data has provided positive evidence for flooding and shown a positive correlation with meteorological data, further studies will be conducted using larger time series. Given the positive evidence for flooding from Twitter data and its positive correlation with meteorological data, it is necessary to conduct further studies with larger time series. Additionally, there are several areas that can be explored further. For example, we could apply supervised machine learning models to develop more accurate alert systems and flood forecasting tools. Additionally, by integrating meteorological data with social network data, we can better understand the spatial patterns of flooding and develop more effective mitigation strategies.”

Reviewer 3 Report

Line 29: An EWS comprises

Line 65: …”The region comprising the lower part of the Tamanduateí river basin …”

Figure 1: The dark blue station marker is nearly invisible against the background. Choose a different color that stands out. The yellow dashed boundary line is not described, or identified in any way.

Author Response

—————————————————

Dear Associate Editor and Reviewers, 

We would like to thank the editor and the reviewers for all their contributions towards improving our manuscript.

The adjustments in the manuscripts are highlighted according to the suggestions provided by Reviewer #1 (red), Reviewer #2 (blue) and Reviewer #3 (magenta).

We believe these improvements suffice to reach all the expectations and to provide high quality discussions in the area.

Yours sincerely,

The authors

—————————————————

Q1) Line 29: An EWS comprises

R: Thank you for your accurate review. We corrected the word in the manuscript. (Please, see lines 29–30).

Q2) Line 65: …”The region comprising the lower part of the Tamanduateí river basin …”

R: Thank you for this correction, we adjusted the text. (Please, see lines 76–79)

Q3) Figure 1: The dark blue station marker is nearly invisible against the background. Choose a different color that stands out. The yellow dashed boundary line is not described, or identified in any way.

R: Many thanks for this suggestion. We revised the study area map location (see Figure 1). (i) The color of the 857A Rain Gauge has been changed to green; (ii) The yellow dashed boundary was properly identified in the revised map (it represents a 2 km buffered area from the rain gauges into the study area and is limited to the Tamanduatei basin region. 

Round 2

Reviewer 1 Report

The manuscript has been markedly improved and all my questions and concerns have been adequately answered. The new figure of the time-series graphs is a useful addition to the MS. Congratulations to the authors! Only slight text editing is needed, minor typos are left in the text (e.g.: line 218, capitalization is needed).

Reviewer 2 Report

The author's responses to the reviewers have been checked. The author has explicitly responded.